# Point-based Correspondence Estimation for Cloth Alignment and Manipulation

Mansi Agarwal, Thomas Weng, David Held

The Robotics Institute, Carnegie Mellon University, USA

{magarwa2,tweng,dheld}@andrew.cmu.edu

*Abstract*—Automating cloth folding is a challenging task with practical implications in various domains. Existing methods often struggle with unaligned configurations, limiting their applicability in real-world scenarios. In this research, we present FabricFlowAlignNet (FFAN), a novel approach that learns flow-based correspondences on point clouds between the current observed and goal cloth configurations. We use these learned 3D correspondences for both cloth alignment and manipulation: correspondences are used to align the observed cloth with the goal, and the flow-based correspondences are re-used as action proposals. Our experiments demonstrate that FFAN demonstrates superior performance compared to a state-of-the-art folding approach, particularly in scenarios where observed cloth is rotated or otherwise unaligned with the goal.

## I. INTRODUCTION

Cloth manipulation is a challenging task, with difficulties in both perception and control due to the deformability of cloth. Manual cloth manipulation techniques are time-consuming, labor-intensive, and prone to human error. As a result, there is a growing demand to automate cloth manipulation in various domains such as folding laundry, handling textiles in manufacturing, and assistive dressing.

A fundamental aspect of successful cloth manipulation is establishing correspondences between the current observation and the goal configuration. These correspondences provide spatial associations necessary for planning and executing folding actions. However, while prior methods have proposed to learn correspondences for cloth [10, 4], they do not explicitly use such methods for reasoning about the *alignment* between the observed cloth and the desired configuration. Alignment is a crucial step in cloth manipulation, and prior correspondence-based policies do not handle cases where the cloth and goal are not aligned [10], or rely on human demonstrations [4].

In this work, we propose **FabricFlowAlignNet** (**FFAN**), an approach that combines the use of correspondences and symmetry-handling techniques to learn a goal-conditioned cloth manipulation policy. Our method leverages correspondences to "virtually" align the observation and goal point clouds, enabling the policy to determine the appropriate actions to execute on the observation. By incorporating these correspondences and symmetry handling, our approach aims to acquire an understanding of cloth folding strategies and develop a manipulation policy capable of accurately and efficiently folding clothes. This is particularly beneficial in challenging scenarios where the observed cloth is rotated or unaligned with the desired goal configuration.

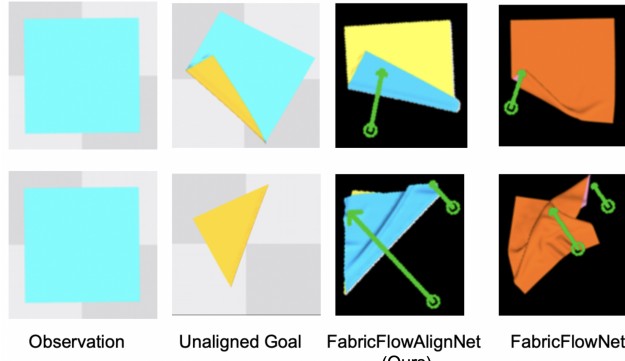

Fig. 1: Performance of FabricFlowAlignNet (FFAN) vs. FNN on unaligned goals. FFAN uses an alignment procedure on learned correspondences to achieve the desired manipulations.

We evaluate the performance of our method against a state-of-the-art folding approach [10] on a folding task, where the goal and observation poses are not aligned. Our method reasons about symmetries and employs correspondences to deal with unaligned goals, unlike the baseline. The results demonstrate the effectiveness and robustness of our approach in achieving successful cloth folding when the observation and goal configurations are unaligned.

## II. PRIOR WORK

FabricFlowNet (FFN) [10] performed bimanual cloth folding by estimating flow correspondences between the observed cloth image and goal cloth image. However, FFN relies on strict alignment between observation and goal cloth poses in the image. Our approach extends FFN by proposing an approach for aligning learned 3D correspondences to overcome these limitations. By establishing spatial relationships between points in observation and goal configurations, we enable precise alignment and achieve better folding performance for unaligned goals than FFN.

Fabric Descriptors [4] is a method for learning correspondences in fabric manipulation tasks using a dense contrastive loss. However, once the correspondences are learned, the proposed policy relies on human demonstrations. In contrast, our method can learn and estimate correspondences without any human demonstrations.

SpeedFolding [1], employs self-supervised learning and a small number of expert demonstrations to perform cloth smoothing and folding. However, Speedfolding is trained

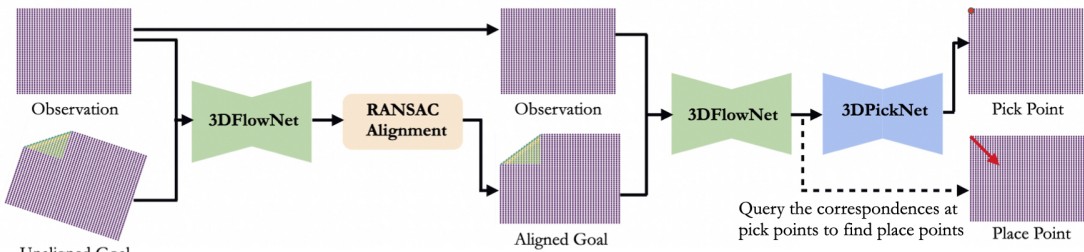

Fig. 2: Overview of the FFAN pipeline.

exclusively in the real world. In comparison, our approach, similar to FFN, undergoes training in simulation before being transferred to the real world.

Cloth Funnels [2] proposes a method that uses self-supervised rewards to learn both cloth canonicalization and alignment. Their alignment procedure is an iterative version of the Procrustes' algorithm, which is designed for aligning rigid objects. However, since the objects being aligned are deformed fabrics, the alignment achieved using Procrustes can be a local optimum. In contrast, our approach proposes using random sample consensus (RANSAC) for aligning deformed fabrics, resulting in an asymptotic, globally optimal alignment.

## III. POINT CLOUD CORRESPONDENCE ESTIMATION FOR CLOTH ALIGNMENT AND MANIPULATION

In this section, we describe FabricFlowAlignNet (FFAN), our approach for estimating observation-goal correspondences to align and manipulate cloth. A schematic overview can be found in Fig. 2.

### A. Learning Correspondences for Point Clouds

As the first component of our overall pipeline, we propose a 3D, flow-based correspondence estimator called "3DFlowNet" (Fig. 3). 3DFlowNet takes the observation and goal point clouds $c_o$ and $c_g$ as input, and outputs 3D flow $\hat{f}$. 3DFlowNet is a non-trivial extension of the FlowNet from FabricFlowNet [10], which was limited to 2D.

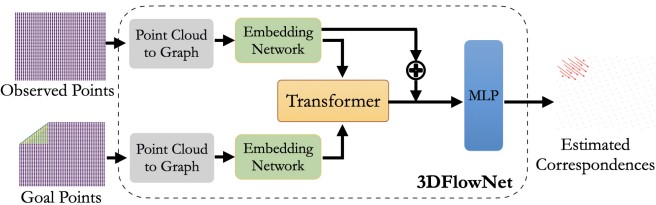

Fig. 3: 3DFlowNet Architecture

We first transform the point clouds into a graph, where nodes represent cloth particles and are connected to their neighboring particles on the cloth mesh. This step requires privileged state information from the simulator of the cloth mesh edges, which would not be available in the real world; estimating these edges is an area of future work and could leverage prior methods like VCD [5]. We embed the input graphs by employing a graph neural network $H$, which outputs embeddings for each node in the graph: $c'_o, c'_g \in \mathbb{R}^{N \times F}$.

We then use a Transformer network [7] denoted as $T$ to perform cross-attention between observation and goal features. Our approach is inspired by prior Transformer-based per-point networks like DCP [9] and TAX-Pose [6]. $T$ takes $c'_o$ and $c'_g$ as input and outputs transformed embeddings $c' \in \mathbb{R}^{N \times F}$. The resulting transformer embeddings, $c'$, are then summed with the original observation embeddings $c'_o$ to produce $c''_o$.

To estimate correspondences, we pass $c''_o$ through MLP layers $M$ to produce estimated correspondences $\hat{f} \in \mathbb{R}^{N \times 3}$. These correspondences represent how each cloth particle transports to achieve the goal configuration.

To train 3DFlowNet, we use ground truth correspondence between point clouds $c_g$ and $c_o$ and a weighted L2 loss $\mathcal{L}_2(\hat{f}, f) = \sum_{i=1}^{N} w_i (f_i - \hat{f}_i)^2$ where $\hat{f}$ represents the estimated correspondences, $f$ represents the ground truth correspondences, and $N$ is the total number of points in the point cloud. The weights $w_i$ are higher for ground truth pick points.

### B. Iterative Correspondence Estimation

To improve correspondence estimation when the displacement between observed and desired goal configurations is large, we introduce an iterative approach to improve the accuracy of our correspondence estimation. Our iterative process involves transporting the input point cloud to the positions indicated by the estimated correspondences, and then re-computing the estimation with this intermediate point cloud. Each iteration of this procedure should further refine the estimated correspondence.

In each iteration, we utilize the trained 3DFlowNet model to estimate the correspondence between the intermediate point cloud $\hat{c}_o$ and the target configuration $c_g$. By integrating the estimated correspondence into the observation, we simulate the application of the flow to progressively approach the target configuration. The algorithm for iterative correspondence estimation is summarized in Alg. 1.

### C. RANSAC Alignment for Unaligned Goals

The correspondences estimated by 3DFlowNet indicate how each point in the observed cloth configuration should move to reach the desired configuration. In the case where the observation and goal are aligned with respect to each other, this per-point flow correspondence represents a desired cloth manipulation, which we can use to estimate the action (Sec. III-D). However, in cases where the observation and goal are not

**Algorithm 1** Iterative Correspondence Estimation
___
1: Input: Trained 3DFlowNet, Point Clouds $c_o, c_g$
2: Initialize all zeros $\bar{f} \in \mathbb{R}^{N \times 3}$
3: $\hat{c}_o := c_o$
4: **for** k = 1...K **do**
5:      $\hat{f} = \text{3DFlowNet}(\hat{c}_o, c_g)$
6:      $\bar{f}$ += $\hat{f}$
7:      $\hat{c}_o$ += $\hat{f}$
8: **end for**
9: **return** $\bar{f}$
___

aligned, the flow correspondences contain both information about alignment as well as the desired manipulation.

To address the cases where the goal is not aligned, we first propose estimating the alignment using the flow-based correspondence and RANSAC [3]. The forward pass through 3DFlowNet provides the estimated correspondences. The RANSAC procedure attempts to find an alignment transform with the maximum number of inlier cloth points as follows:

1) Sample three indices $(i, j, k)$ on the cloth.
2) Compute the transformation matrix $T$ between the 3 sampled cloth points $(p_i, p_j, p_k)$ and their estimated correspondences $(p_i + \hat{f}_i, p_j + \hat{f}_j, p_k + \hat{f}_k)$.
3) Compute inliers by transforming all current cloth points $p$ according to $T$, computing the distance between transformed points and points transported using estimated flow $||Tp - (p + \hat{f})||$, and thresholding the per-point distance by an epsilon $\epsilon$.
4) Sample $m$ times and choose the transformation matrix $T$ with the maximum number of inliers.

Once the alignment $T$ has been estimated, we virtually align the observation and goal, re-estimate correspondences given the alignment, and determine the manipulation action according to the following section.

### D. Estimating the Pick Location for an Action

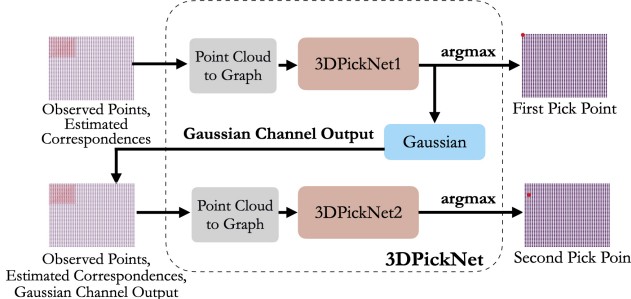

Fig. 4: 3DPickNet Architecture

To predict the pick points necessary for cloth manipulation, we introduce a neural network called "3DPickNet". Similar to FFN [10], our method supports bimanual manipulation and is capable of estimating both pick points $p_1$ and $p_2$. The inputs to 3DPickNet are the current observation $c_o$ and the estimated

correspondences $\hat{f}$ between $c_o$ and the goal configuration $c_g$. The architecture of 3DPickNet is depicted in Figure 4.

To enable the prediction of the second pick point conditioned on the first pick point, we utilize two separate networks: 3DPickNet1 and 3DPickNet2. In 3DPickNet1, we concatenate $c_o$ and $\hat{f}$ and create a graph representation of the point cloud. Each node in the graph is represented as $[x, y, z, \hat{f}]$. 3DPickNet1 generates a probability value for each node to be selected as the first pick point $p_1$. The node with the highest probability is identified as $p_1$.

3DPickNet2 is responsible for predicting the second pick point $p_2$, taking $p_1$ into account. In this network, we introduce an additional input channel called $\hat{p}_1$, which represents a 3D Gaussian distribution centered on $p_1$. This channel assigns higher values to nodes near $p_1$ and lower values to nodes farther away, to give PickNet2 information about the first pick location when predicting the second pick point $p_2$.

For training 3DPickNet, we use a weighted binary cross-entropy loss. The loss function compares the predicted probabilities of nodes being pick points with the ground truth labels. The binary cross-entropy loss function is defined as:

$$L(p, y) = \sum_{i=1}^{N} -w_i(y_i \log p_i + (1 - y_i) \log(1 - p_i)) \quad (1)$$

where $p$ represents the predicted probabilities, $y$ is the ground truth labels, and $N$ is the total number of nodes. The weights $w_i$ are higher for ground truth pick points.

Once the pick points $p_1$ and $p_2$ are predicted using the estimated correspondences $\hat{f}$, the corresponding actions can be executed to achieve the desired goal configuration.

### E. Implementation Details

We use the same dataset as FabricFlowNet [10], but use point clouds of the cloth instead of depth images, and use 3D pick and place points instead of 2D. The graph neural network $H$ for 3DFlowNet consists of two Graph Attention layers (GATConv) [8]. The MLP network architecture $M$ consists of two fully-connected layers. The 3DPickNet architecture consists of three Graph Attention Network layers and two MLP layers for both 3DPickNet1 and 3DPickNet2. At the end of each network, a Sigmoid layer computes the probability of each node being a pick point. 3DPickNet1 represents each node with a six-dimensional feature, while 3DPickNet2 utilizes a seven-dimensional feature to accommodate the additional information provided by $\hat{p}_1$.

## IV. EXPERIMENTS

Our experiments investigate the following questions: (1) How does FFAN compare with FabricFlowNet (FFN) [10] on aligned goals? (2) How does FFAN compare with FFN on unaligned goals? We evaluate the methods in simulation, using the average L2 distance between cloth points in the achieved vs. desired point clouds as our error metric.

### A. Performance on Aligned Goals

We use the same test set as FFN [10] to evaluate performance on aligned goals. This test set consists of 40 single-step goals, where both the observation and the goal positioned at the center of the workspace with the same orientation. For this experiment, we do not use alignment estimation with FFAN to directly compare pre-aligned folding performance.

Table I presents the performance comparison between our method and FFN. The results demonstrate that our method performs comparably to FFN on aligned goals, with only a marginal difference in average particle distance.

TABLE I: Folding Performance on 40 Aligned Goals

| Method | Average Particle Distance (mm) ↓ |
| --- | --- |
| FFN [10] | 4.26 |
| FFAN (Ours) | 5.54 |

### B. Performance on Unaligned Goals

We also evaluate the performance on unaligned goals, where the goal cloth configuration is randomly rotated and therefore not aligned with the initial observed configuration. We conducted experiments on three test sets: Easy, Medium, and Hard, where each test set corresponds to a different range of rotations. Easy encompasses angles between -5 and 5 degrees, Medium ranges from -45 to 45 degrees, and Hard covers a complete rotation from 0 to 360 degrees.

We evaluated the performance of FFAN in two scenarios: using ground truth vs. estimated correspondences for RANSAC alignment. Figure 5 presents a comparison of the two methods against FFN across all four test sets: aligned, easy, medium, and hard. From the results, we observe that our method with estimated correspondences outperforms FFN on the Medium and Hard tasks. However, using ground truth correspondences for RANSAC alignment yields even better results across all misaligned sets, surpassing the performance of FFN. This demonstrates the potential for further improvement by improving the correspondence estimation. Qualitative results on the Medium case can be found in Fig. 1.

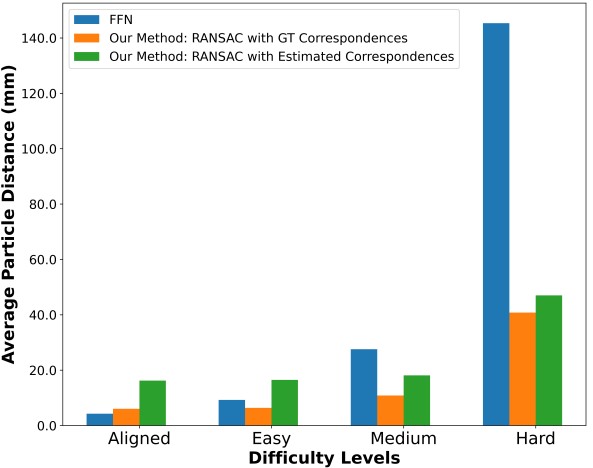

Fig. 5: Comparison of Folding on Different Test Sets

### C. Ablations

*1) No Iterative Correspondence:* In this section, we ablate our approach by removing iterative correspondence estimation (Sec. III-B). Table II shows that average particle distance error is higher when iterative correspondence estimation is removed.

TABLE II: Ablation of Iterative Correspondence Estimation

| Method | Average Particle Distance (mm) ↓ |
| --- | --- |
| FFAN w/o Iter. Corresp. | 10.591 |
| FFAN w/ Iter. Corresp. | **5.54** |

*2) Number of Iterations for Iterative Correspondence:* To determine the number of iterations to run for iterative correspondence estimation, we measured performance while increasing the number of iterations on a validation set. We used flow prediction error, an unweighted version of the loss from Sec. III-A , as our performance metric. We evaluated number of iterations $k = 1$ (run 3DFlowNet once) to $4$. Note that we did not retrain 3DFlowNet in an iterative manner.

Figure 6 shows the flow prediction error as a function of the number of iterations ($k$) in the iterative flow process. As we increase $k$ from 1 to 3, there is a notable decrease in the flow prediction error; however, beyond $k = 3$, we observed a slight increase in the error. Based on these observations, we empirically determined that the number of iterations for iterative flow correspondence estimation is $k = 3$.

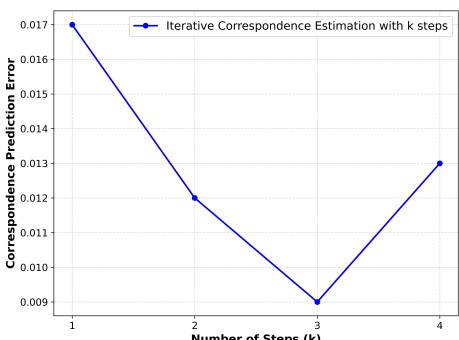

Fig. 6: Error vs. Number of Correspondence Estimation Steps

## V. CONCLUSION

In this work, we propose **FabricFlowAlignNet** (**FFAN**), a goal-conditioned policy for cloth alignment and folding. Our approach estimates flow correspondences to reason about the alignment between the observed cloth and desired goal, then predicts actions given the estimated alignment. FFAN performs on par with FFN for aligned goals, and outperforms FFN when handling large misalignments. Our ablations demonstrate the importance of using iterative correspondence estimation and of selecting the number of iterations.

**Limitations and Challenges**: Our method also currently requires meshes as input; for cloth manipulation in the real world, such a mesh will have to be estimated. Like FFN, our method relies on sub-goals, which can be restrictive and may not generalize well to unseen fabrics and configurations. Exploring alternative approaches that eliminate explicit sub-goals is a potential direction for future work.

ACKNOWLEDGMENTS

This work was supported by the US Air Force and DARPA (FA8750-18-C-0092) and the NSF (IIS-1849154, DGE2140739).

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
