# OpenReview forum: "Point-based Correspondence Estimation for Cloth Alignment and Manipulation"
_roboticsfoundation.org/RSS/2023/Workshop/Symmetry — RSS 2023 Workshop Symmetry_

### Official Review · Reviewer_q6zP · 2023-06-16

**Rating:** 7
**Confidence:** 4

**Review:**

The paper proposes FabricFlowAlignNet, a novel approach for learning goal-conditioned cloth manipulation policies from point cloud inputs. The method contains three main components. First, a GAT+Transformer architecture estimates the correspondence between the observation and the goal. Second, a RANSAC-based method aligns the observation and the goal using the correspondence and re-estimates the correspondence after the alignment. Last, two pick networks, taking in both the point cloud and the estimated correspondence, generate two pick points for a bi-manual manipulation policy.

The proposed method demonstrates a compelling experimental result, especially when the goal and the observation are not aligned (i.e., there is a rotation or translation between the fabrics in the observation and the goal). The paper is well-written and easy to follow, however, I would suggest the authors include a figure for the entire architecture and information flow, ideally combining Fig.2 and Fig.3 with the additional RANSAC component, to better convey the high-level idea.

As acknowledged by the authors, a limitation of the work is that implementing the approach in the real world is a bit challenging (even just to evaluate the learned policy). Another consideration is that the experiments do not seem to underscore the advantages (aside from the ability to perform alignment using RANSAC) of using the proposed point cloud based method vs the image based method in the prior work [11]. For future research, I would also recommend the authors consider the use of SE(2) or SE(3) equivariant neural network architectures (e.g., e3nn, vector neuron, etc.), which might be especially effective in solving the alignment issue.

---

### Official Review · Reviewer_VpGr · 2023-06-16
**Good and interesting paper.**

**Rating:** 7
**Confidence:** 4

**Review:**

Summary:
The paper proposes a novel point-based method for cloth manipulation. A 3D flow estimation network and a 3D action predictor network are employed together to conduct goal-conditioned cloth manipulation. Additionally, RANSAC and an iterative estimation technique are used to align the points and improve accuracy. The authors demonstrate the method's potential in simulation and empirically analyze its performance on tasks of varying difficulty levels.

Comments:
1. The alignment part is a practical incremental enhancement of FabricFlowNet. This part implicitly uses symmetries to align the observation and goal. To resonate more with the topic of this workshop, authors could explicitly apply symmetry constraints into the neural network, e.g., rotations or reflections, considering the graph/point representation has the potential to do so, as shown in the past literature.

2. While the simulation results have demonstrated the method's effectiveness,  it is crucial to incorporate real-world experiments in this paper for 2 reasons: 1) Sim-to-real gap: given that estimating an accurate graph of a real physical cloth could be challenging; 2) As an enhancement over a previous paper, simulation experiments are not sufficient since FabricFlowNet has real-world experiments. Without privileged information from the simulator, the inaccurately estimated graph might lead to a very different performance in the real world. Given that the authors mentioned VCD can be used for edge estimation, it would be much stronger if the authors can evaluate their method with VCD in physical experiments.

3. The paper is overall clear and comprehensive.  A suggestion for improvement is to include an overview figure illustrating all sub-modules such as 3DPickNet, 3D FlowNet, RANSAC within a single FabricFlowAlignNet figure. This will help readers better comprehend the overall process and the associated concepts, particularly at an initial read-through.

4. In Table 1, the authors compared their method with FFN and shows comparable performance, which shows the potential of point-based methods. However, better performance over the image-based baseline is expected because 3D points potentially offer more information while images become unobservable due to the self-occlusion (folding) of the cloth. Without such improvement, the motivation for transferring from images to points for cloth manipulation is unclear.

Conclusion:
 The overall framework is based on a precedent work FabricFlowNet with several key modifications and improvements. The point-based method shows its potential to be used on cloth manipulation tasks. However, the paper would benefit from more comprehensive baseline comparisons and sim-to-real experiments to convincingly highlight the advantages of the proposed 3D method over previous image-based approaches. Overall, I incline to accept this paper.

---

### Decision · Program_Chairs · 2023-06-23

**Decision:**

Accept

**Comment:**

Congratulations! We encourage the authors to revise the paper based on the reviewer's feedback.
Your paper will be presented as both a short presentation and a poster. Detailed instructions about the presentation format and camera-ready submission will be sent to you soon.